# What Do You See?
# Enhancing Zero-Shot Image Classification with Multimodal Large Language Models

## Abstract

Large language models (LLMs) have been effectively used for many computer vision tasks, including image classification. In this paper, we present a simple yet effective approach for zero-shot image classification using multimodal LLMs. By employing multimodal LLMs, we generate comprehensive textual representations from input images. These textual representations are then utilized to generate fixed-dimensional features in a cross-modal embedding space. Subsequently, these features are fused together to perform zero-shot classification using a linear classifier. Our method does not require prompt engineering for each dataset; instead, we use a single, straightforward, set of prompts across all datasets. We evaluated our method on several datasets, and our results demonstrate its remarkable effectiveness, surpassing benchmark accuracy on multiple datasets. On average over ten benchmarks, our method achieved an accuracy gain of 4.1 percentage points, with an increase of 6.8 percentage points on the ImageNet dataset, compared to prior methods. Our findings highlight the potential of multimodal LLMs to enhance computer vision tasks such as zero-shot image classification, offering a significant improvement over traditional methods.

## 1 Introduction

Zero-shot image classification aims to categorize images into classes unseen during training, presenting a significant challenge in computer vision. Recent approaches leverage the power of large language models (LLMs) like GPT-4 (Brown et al. (2020)) to generate prompts for target classes, often in conjunction with vision-language models such as CLIP (Radford et al. (2021)) to embed images and text in a common space. Open-vocabulary models like CLIP (Radford et al. (2021)) and VirTex (Desai & Johnson (2021)) have shown promise in this area due to their ability to generalize to unseen classes. These models learn to match images with captions from vast amounts of image-text data, allowing for dynamic classification without retraining. Early works like DeViSE (Frome et al. (2013)) pioneered the concept of a joint embedding space for images and text, enabling generalization to unseen classes. Approaches like CLIP (Radford et al. (2021)), based on contrastive learning, and ALIGN (Jia et al. (2021)), employing a two-stage framework, further refined the alignment of image and text representations. More recent approaches for zero-shot image classification (e.g., Pratt et al. (2023); Menon & Vondrick (2023)) have utilized LLMs to generate prompts (i.e., captions or descriptions) for the target classes to further improve the classification accuracy.

However, relying solely on visual features of the input images during inference can limit accuracy, as these features may not be sufficient to fully capture the nuances present in textual descriptions. To address this, we propose a novel method that leverages the capabilities of *multimodal* LLMs to generate rich textual representations of the input images. Multimodal LLMs, such as GPT-4 (Brown et al. (2020)) and Gemini (Gemini Team Google (2023)), have demonstrated remarkable abilities in various tasks. They are capable of processing and integrating information from various sources like text, images, and audio. This allows them to perform tasks that were previously impossible, such as generating detailed image descriptions, answering complex visual questions, and even creating realistic images from text. Inspired by these advancements, we utilize a straightforward set of prompts to generate detailed textual descriptions of the input images, eliminating the need for complex prompt engineering seen in previous works (e.g., Radford et al. (2021); Guo et al. (2023)).

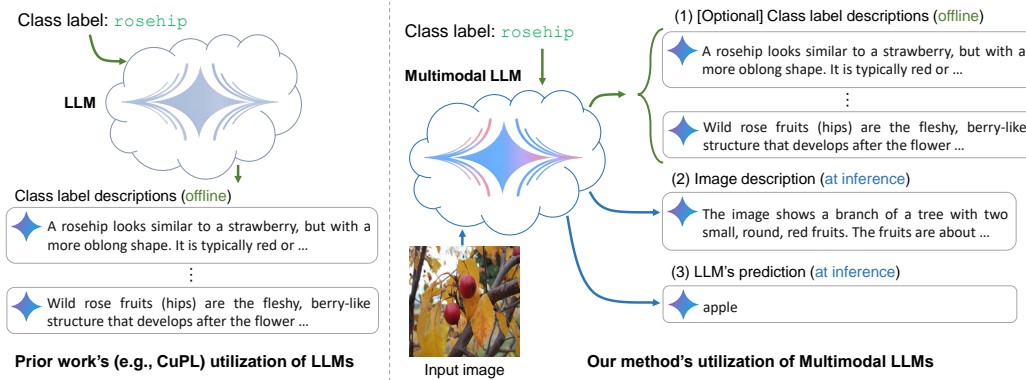

Figure 1: An illustration of the difference between our work and prior work (e.g., CuPL (Pratt et al. (2023))) in terms of using LLMs. Prior works use LLMs to describe class labels while we use multimodal LLMs to describe input images and class labels as well as making initial predictions. The shown image is sourced from the ImageNet dataset (Deng et al. (2009)).

These textual representations are then fused with visual features to perform zero-shot classification. See Figure 1.

Our method offers several key advantages: it significantly improves classification accuracy by incorporating richer textual information extracted directly from the input images; it employs a simple and universal set of prompts, eliminating the need for dataset-specific prompt engineering; and it outperforms existing methods on a variety of benchmark datasets. By employing multimodal LLMs and a straightforward set of prompts, our method outperforms previous zero-shot image classification methods. Specifically, our method achieves an average accuracy gain of 4.1 percentage points over ten image classification benchmark datasets and an accuracy increase of 6.8% on the ImageNet dataset (Deng et al. (2009)).

In the following sections, we detail our proposed approach for zero-shot image classification using LLMs (Section 2), present experimental results across ten benchmark datasets (Section 3), analyze the computational resources used (Section 4), discuss limitations (Section 5), and conclude with future directions (Section 6).

## 2 METHOD

Given an input image, $\mathbf{X}$, containing object(s) belonging to a single class label from a finite set of class labels, $\{l_i\}_{i=1}^m$, our objective is to classify $\mathbf{X}$ without any dataset-specific training process for image classification. The overview of our method is illustrated in Figure 2. As shown in Figure 2, our approach relies on a cross-modal embedding encoder models (image encoder, $f_i$, and text encoder, $f_t$), trained to learn joint representations of images and text, as demonstrated in prior works, such as Radford et al. (2021); Frome et al. (2013); Jia et al. (2021). Additionally, we utilize a multimodal LLM, $g$, which is pre-trained on a large corpus of multimodal data. This model, $g$, is designed to generate responses that align with both textual and visual inputs, effectively integrating information from both modalities for enhanced predictions (e.g., Gemini Team Google (2023)).

### 2.1 CLASS LABEL FEATURES

Our zero-shot classifier utilizes class label features of the target dataset. We first encode the set of class labels, $\{l_i\}_{i=1}^m$, in some embedding space by the cross-modal encoder models, $f_i$, and $f_t$, such that $\{\mathbf{L}_i\}_{i=1}^m$ refers to the set of encoded class label features, where $\mathbf{L}_i \in \mathbb{R}^n$ is a normalized $n$-D class label feature of the class label $l_i$ and $n$ is the dimensionality of the embedded features.

Our zero-shot classifier uses the embedded class labels by representing $\{\mathbf{L}_i\}$ as a 2D matrix, $\mathbf{M} \in \mathbb{R}^{n \times m}$, by stacking all encoded class label features:

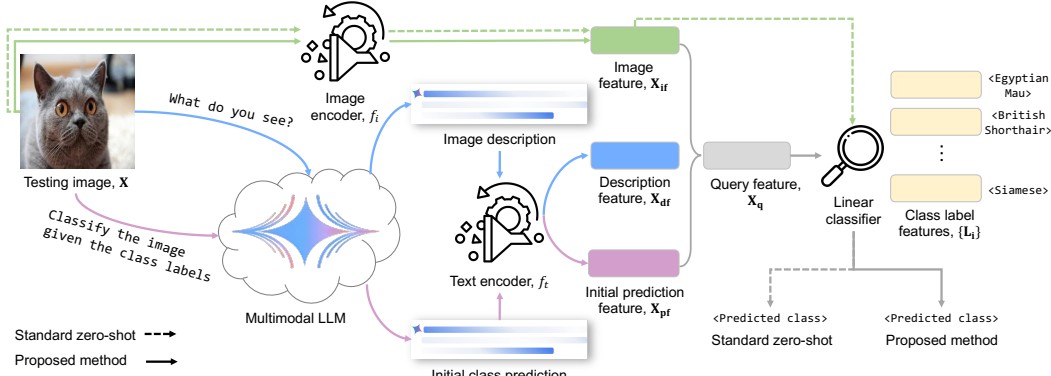

Figure 2: We propose a zero-shot image classification method that leverages multimodal large language models (LLMs) to enhance the accuracy of standard zero-shot classification. Our method employs a set of engineered prompts to generate image description and initial class prediction by the LLM. Subsequently, we encode this data along with the input testing image using a cross-modal embedding encoder to project the inputs into a common feature space. Finally, we fuse the generated features to produce the final query feature, which is then utilized by a standard zero-shot linear image classifier to predict the final class. The shown image is sourced from the Pets dataset (Parkhi et al. (2012)).

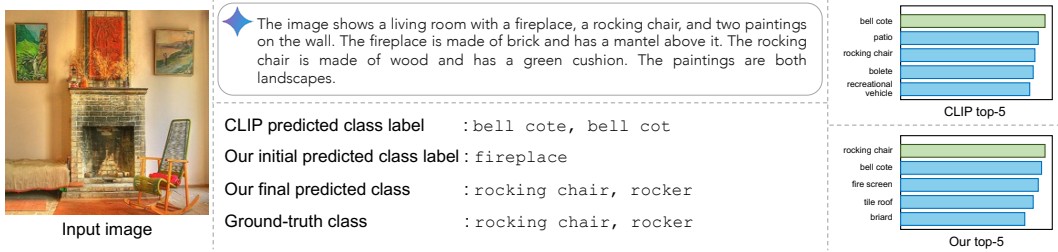

Figure 3: Our method utilizes image description and initial class prediction generated by LLM, in addition to the input image, to improve the zero-shot classification accuracy of cross-modal embedding models, such as CLIP (Radford et al. (2021)). The shown image is from the ImageNet dataset (Deng et al. (2009)).

$$\mathbf{M} = [\mathbf{L}_1, \ldots, \mathbf{L}_m]. \tag{1}$$

Such an encoded feature matrix can be generated using one of three options: (1) directly from the textual class labels, (2) using a human-designed template – e.g., ``A photo of {class_label}'' (Radford et al. (2021); Guo et al. (2023)), where {class_label} refers to the textual label of one of our classes $\{l_i\}_{i=1}^m$, or (3) LLM-generated class description(s) (Pratt et al. (2023)). In option (3), the LLM-generated class description(s) are then converted to embedded features, followed by fusion (e.g., averaging) to generate a single embedded feature for each class label in the dataset. Optionally, all features from the three options can be fused together (e.g., averaged) for increased robustness.

## 2.2 CROSS-MODAL INPUT FEATURES

To predict the final class, we first encode the input image by the cross-modal image encoder model, $f_i$, to generate the image feature $\mathbf{X}_{if} \in \mathbb{R}^n$ as described below:

$$\widetilde{\mathbf{X}}_{\mathtt{if}} = f_i\left(\mathbf{X}\right), \tag{2}$$

$$\mathbf{X}_{\mathtt{if}} = \frac{1}{\|\widetilde{\mathbf{X}}_{\mathtt{if}}\|}\widetilde{\mathbf{X}}_{\mathtt{if}}, \tag{3}$$

where $\|\cdot\|$ performs vector normalization. Traditionally, the image feature serves as the sole input to prior zero-shot image classifiers (Pratt et al. (2023); Radford et al. (2021); Guo et al. (2023)). Our method enhances this input by incorporating the LLM, $g$, which generates additional textual-based inputs for our zero-shot image classifier (see Figure 3). To achieve this, we employ an engineered prompt that instructs the LLM to describe the input image, $\mathbf{X}$, and perform initial image classification using the textual names of the class label set $\{l_i\}_{i=1}^{m}$. Denoting $p_{\mathtt{d}}$ and $p_{\mathtt{c}}$ as our prompts for image description and initial image classification, respectively, we generate two additional embedded features alongside $\mathbf{X}_{\mathtt{if}}$ by:

$$\widetilde{\mathbf{X}}_{\mathtt{df}} = (f_t \circ g)\left(\mathbf{X}, p_{\mathtt{d}}\right), \tag{4}$$

$$\mathbf{X}_{\mathtt{df}} = \frac{1}{\|\widetilde{\mathbf{X}}_{\mathtt{df}}\|}\widetilde{\mathbf{X}}_{\mathtt{df}}, \tag{5}$$

$$\widetilde{\mathbf{X}}_{\mathtt{pf}} = (f_t \circ g)\left(\mathbf{X}, p_{\mathtt{c}}\right), \tag{6}$$

$$\mathbf{X}_{\mathtt{pf}} = \frac{1}{\|\widetilde{\mathbf{X}}_{\mathtt{pf}}\|}\widetilde{\mathbf{X}}_{\mathtt{pf}}, \tag{7}$$

where $\mathbf{X}_{\mathtt{df}} \in \mathbb{R}^n$ and $\mathbf{X}_{\mathtt{pf}} \in \mathbb{R}^n$ refer to the image description feature and initial class prediction feature, respectively. Notably, in our method, unlike prior methods (e.g., Pratt et al. (2023); Radford et al. (2021); Guo et al. (2023)), such prompts $p_{\mathtt{d}}$ and $p_{\mathtt{c}}$ do not require dataset-specific engineering for each dataset. Instead, we employ fixed prompts: the first prompt instructs the LLM to provide a generic image description, while the second prompt includes the textual class labels of the target dataset. Further details regarding our prompts are provided in the supplemental materials (Appendix A).

After generating the three input features (image feature, description feature, and initial prediction feature), we fuse them to generate our final query feature, $\mathbf{X}_{\mathtt{q}}$. One can interpret this fusion as an ensemble of different candidate features to generate a more precise query feature (see Figure 4). We adopted a simple fusion, where the final query feature, $\mathbf{X}_{\mathtt{q}}$, is generated by:

$$\widetilde{\mathbf{X}}_{\mathtt{q}} = \mathbf{X}_{\mathtt{if}} + \mathbf{X}_{\mathtt{df}} + \mathbf{X}_{\mathtt{pf}}, \tag{8}$$

$$\mathbf{X}_{\mathtt{q}} = \frac{1}{\|\widetilde{\mathbf{X}}_{\mathtt{q}}\|}\widetilde{\mathbf{X}}_{\mathtt{q}}. \tag{9}$$

We found that this simple fusion yields good results compared to alternative fusion approaches. Refer to Section 3.2 for ablation studies.

## 2.3 CLASS LABEL PREDICTION

After computing our query feature, $\mathbf{X}_{\mathtt{q}}$, we apply our zero-shot linear classifier weights, $\mathbf{M}$, to the fused query feature, $\mathbf{X}_{\mathtt{q}}$, to generate the final similarity scores (i.e., "logits") of our prediction. This process can be described as follows:

$$\mathbf{W} = \mathbf{X}_{\mathtt{q}}^{T}\mathbf{M}, \tag{10}$$

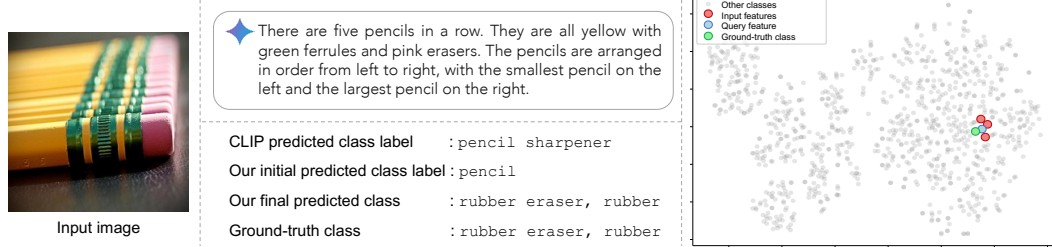

Figure 4: Our query feature is a fusion of features extracted from the input image, image description, and initial prediction. This fusion operates similarly to ensembling, where our fused query feature demonstrates better robustness, achieving higher accuracy compared to traditional image features used in cross-modal-based zero-shot image classification; e.g., CLIP (Radford et al. (2021)). On the right, the t-SNE (Van der Maaten & Hinton (2008)) plot shows the class-embedded features of the ImageNet dataset (Deng et al. (2009)) (in gray, with the ground-truth class of the shown image in green), our input features (in red), and the query feature after fusion (in blue).

where $T$ represents the vector transpose operation to transform $\mathbf{X}_q$ into a row vector of shape $1 \times n$, and $\mathbf{W} \in \mathbb{R}^{1 \times m}$ contains the similarity scores of the generated query feature to the class label features in the target dataset. The index of the final predicted class is then computed as $\mathrm{argmax}(\mathbf{W})$ that corresponds to the maximum similarity score.

## 3 EXPERIMENTS

In our experiments, we employed Gemini Pro (Gemini Team Google (2023)) as our multimodal LLM, $g$, for generating image descriptions and initial predictions. We utilized CLIP (ViT-L/14) (Radford et al. (2021)) as our cross-modal embedding encoder models, $f_i$ and $f_t$, to encode input testing images, image descriptions, and initial predictions generated by Gemini Pro.

We explored four different versions of class label features, $\mathbf{M}$, in Equation 1. Specifically, we used CLIP to encode the following representations of class labels: 1) class label names, 2) the text template ``A photo of {class_label}'', where {class_label} denotes each class label in each dataset, and 3) class descriptions, similar to those generated by CuPL (Pratt et al. (2023)), and 4) a combination of the aforementioned three options—akin to the fusion of our input features, we combined the three encoded features of each class and computed their average feature. The class descriptions were produced by prompting Gemini Pro to describe each class label in the dataset 50 times, resulting in 50 different class descriptions for each class. Subsequently, we utilized CLIP to encode the 50 class descriptions for each class and compute the average encoded feature vector to represent each class label. The class description features are generated once as described in Section 2. The exact prompts used to generate image descriptions, initial predictions, and class description are detailed in Appendix A.

### 3.1 RESULTS

We evaluated our method on the following datasets: ImageNet (Deng et al. (2009)), Pets (Parkhi et al. (2012)), Places365 (Zhou et al. (2017)), Food-101 (Bossard et al. (2014)), SUN397 (Xiao et al. (2010; 2016)), Stanford Cars (Krause et al. (2013)), Describable Textures Dataset (DTD) (Cimpoi et al. (2014)), Caltech-101 (Fei-Fei et al. (2004)), CIFAR-10 (Krizhevsky et al. (2009)), and CIFAR-100 (Krizhevsky et al. (2009)). We compared our results against the following zero-shot classification methods: CLIP (Radford et al. (2021)), SLIP (Mu et al. (2022)), PyramidCLIP (Gao et al. (2022)), nCLIP (Zhou et al. (2023)), NLIP (Huang et al. (2023)), UniCLIP (Lee et al. (2022)), ALIP (Yang et al. (2023)), CALIP (Guo et al. (2023)), and CuPL (Pratt et al. (2023)). For CuPL (Pratt et al. (2023)), we utilized Gemini Pro (Gemini Team Google (2023)) for computing class descriptions instead of GPT-3 (Brown et al. (2020)), ensuring a fair comparison with our method, which also employs Gemini Pro. Additionally, it is worth mentioning that the results reported in

Table 1: Comparison of classification accuracy between our method and prior work across various datasets, including ImageNet (Deng et al. (2009)), CIFAR-10 (C-10) (Krizhevsky et al. (2009)), CIFAR-100 (C-100) (Krizhevsky et al. (2009)), Food-101 (Bossard et al. (2014)), SUN397 (Xiao et al. (2010; 2016)), Cars (Krause et al. (2013)), DTD (Cimpoi et al. (2014)), Caltech-101 (Fei-Fei et al. (2004)), Pets (Parkhi et al. (2012)), and Places (Zhou et al. (2017)). We report our results with the following class label features: 1) class descriptions, 2) class labels, 3) the template "A photo of {class}", and combined features of (1-3). The symbol $*$ denotes training-free methods, whereas the symbol $\diamond$ represents few-shot methods. The best results are highlighted in yellow, and the best zero-shot classification results are highlighted in **bold**.

| | Dataset (number of classes/number of testing images) | | | | | | | | | |
|---|---|---|---|---|---|---|---|---|---|---|
| **Method** | ImageNet (1K/50K) | C-10 (10/10K) | C-100 (100/10K) | Food (101/25.3K) | SUN (397/19.9K) | Cars (196/8K) | DTD (47/1.9K) | Caltech (101/8.7K) | Pets (37/3.7K) | Places (365/36.5K) |
| CLIP (ViT-L/14) (Radford et al. (2021)) | 65.1 | 87.6 | 54.3 | 86.8 | 61.2 | 65.1 | 46.7 | 83.4 | 87.9 | 37.3 |
| CLIP (ViT-B/32) (Radford et al. (2021)) | 47.0 | 73.4 | 41.4 | 70.8 | 56.5 | 43.8 | 37.0 | 84.4 | 72.9 | 35.6 |
| CLIP (ViT-B/16) (Radford et al. (2021)) | 54.9 | 66.5 | 37.6 | 80.1 | 58.5 | 52.6 | 37.9 | 85.3 | 80.4 | 36.3 |
| CLIP (RN50) (Radford et al. (2021)) | 43.0 | 42.7 | 16.0 | 57.4 | 46.3 | 34.3 | 30.5 | 79.1 | 64.7 | 29.7 |
| CLIP (RN101) (Radford et al. (2021)) | 44.0 | 49.0 | 22.4 | 64.3 | 50.2 | 44.5 | 34.0 | 83.2 | 66.3 | 30.9 |
| SLIP (Mu et al. (2022)) | 47.9 | 87.5 | 54.2 | 69.2 | 56.0 | 9.0 | 29.9 | 80.9 | 41.6 | - |
| PyramidCLIP (Gao et al. (2022)) | 47.8 | 81.5 | 53.7 | 67.8 | 65.8 | 65.0 | 47.2 | 81.7 | 83.7 | - |
| nCLIP (Zhou et al. (2023)) | 48.8 | 83.4 | 54.5 | 65.8 | 59.9 | 18.0 | 57.1 | 73.9 | 33.2 | - |
| NLIP (Huang et al. (2023)) | 47.4 | 81.9 | 47.5 | 59.2 | 58.7 | 7.8 | 32.9 | 79.5 | 39.2 | - |
| UniCLIP (Lee et al. (2022)) | 54.2 | 87.8 | 56.5 | 64.6 | 61.1 | 19.5 | 36.6 | 84.0 | 69.2 | - |
| ALIP (Yang et al. (2023)) | 40.3 | 83.8 | 51.9 | 45.4 | 47.8 | 3.4 | 23.2 | 74.1 | 30.7 | - |
| CALIP (ViT-B/32) (Guo et al. (2023)) | 60.6 | 76.5 | 44.2 | 77.4 | 58.6 | 56.3 | 42.4 | 87.7 | 86.2 | 36.9 |
| CALIP (ViT-B/16) (Guo et al. (2023)) | 57.5 | 70.2 | 41.3 | 80.7 | 60.6 | 50.1 | 39.5 | 86.2 | 79.1 | 38.4 |
| CALIP (RN101) (Guo et al. (2023)) | 50.0 | 49.7 | 24.5 | 67.3 | 51.8 | 39.4 | 35.4 | 84.3 | 70.5 | 33.8 |
| CuPL (Pratt et al. (2023)) | 66.6 | 86.6 | 57.7 | 89.0 | 65.3 | 63.9 | 49.1 | 90.5 | 80.0 | 39.7 |
| Tip-Adapter ($*$) (Zhang et al. (2022)) | 62.0 | - | - | - | - | - | - | - | - | - |
| SuS-X ($*$) (Udandarao et al. (2023)) | 61.9 | - | - | - | - | - | 50.6 | - | 77.6 | - |
| Tip-Adapter-F ($\diamond$) (Zhang et al. (2022)) | 65.5 | - | - | - | - | - | - | - | - | - |
| CLIP-Adapter ($\diamond$) (Gao et al. (2024)) | 61.3 | - | - | - | - | - | 66.1 | 93.4 | - | - |
| APE-T ($\diamond$) (Zhu et al. (2023)) | 66.1 | - | - | - | - | - | - | - | - | - |
| Ours (descriptions) | 71.3 | 91.2 | 65.3 | 92.5 | 68.8 | 72.0 | 55.1 | 91.3 | 85.0 | 42.0 |
| Ours (class labels) | 69.9 | 91.8 | 65.4 | 92.0 | 66.8 | 74.3 | 53.2 | 88.5 | 90.0 | 41.3 |
| Ours (template) | 69.0 | 93.1 | 65.8 | 89.3 | 64.5 | 73.6 | 52.3 | 84.5 | 87.9 | 39.4 |
| Ours (combined) | **73.4** | **93.4** | **70.2** | **93.0** | **70.6** | **76.6** | **58.0** | 89.4 | **90.9** | **43.4** |

the original paper of CuPL (Pratt et al. (2023)) use different class labels for the ImageNet dataset (Deng et al. (2009)) compared to the original class labels of the dataset. Consequently, we decided to re-compute the results for CuPL using Gemini Pro with the standard ImageNet class labels.

For CLIP (Radford et al. (2021)) and CALIP (Guo et al. (2023)), we employed the template "A photo of {class_label}" to encode textual class labels across all datasets, except for Pets (Parkhi et al. (2012)), DTD (Cimpoi et al. (2014)), and Cars (Krause et al. (2013)) datasets. For these exceptions, we employed specific templates: "A photo of {class_label}, a type of pets", "A photo of {class_label}, a textural category", and "A photo of {class_label}, a car model", respectively. This approach was found to enhance results, consistent with prior findings (Radford et al. (2021); Li et al. (2023); Allingham et al. (2023); Popp et al. (2024)).

For consistency, we resize all images to 224×224 before processing them with our method, CLIP, CuPL, and CALIP methods. Additionally, we report results from training-free and few-shot learning methods for a comprehensive comparison, including: Tip-Adapter (Zhang et al. (2022)), SuS-X (Udandarao et al. (2023)), CLIP-Adapter (Gao et al. (2024)), and APE-T (Zhu et al. (2023)).

The top-1 accuracy results are reported in Table 1 (see Appendix B for top-5 accuracy results). As can be seen, our method consistently achieves state-of-the-art results when compared with prior work across all datasets in zero-shot image classification methods, and in the majority of datasets when considering other methods (i.e., training-free and few-shot methods). Using the combined class features yields the most promising results across the majority of datasets, with the exception of Caltech-101 (Fei-Fei et al. (2004)), where the best results were achieved using the class description features.

Figure 5 shows visual examples, where we highlight parts of input modalities (image, initial prediction, and image description) based on their contribution to the final predicted class. To emphasize the significance of each input component in the final prediction, we employ a straightforward approach. Specifically, we utilize a 2D sliding kernel that traverses the image, masking out patches of the image. Subsequently, we measure the difference between the initial prediction and the prediction after masking to highlight areas of the image that contribute most significantly to the final prediction. Similarly, we apply this approach to the textual inputs. Given two distinct input texts – namely, the

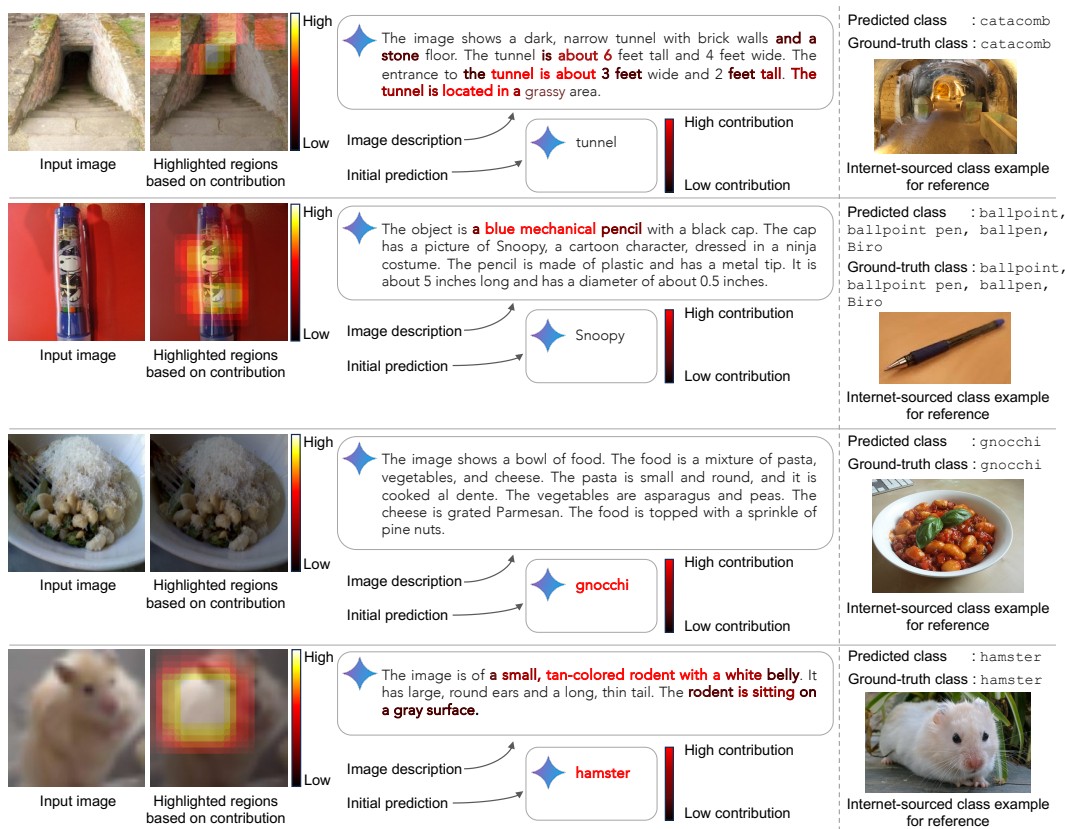

Figure 5: Input data highlighted based on its contribution to the final prediction. Examples are shown from the Places (Zhou et al. (2017)) (first row), ImageNet (Deng et al. (2009)) (second row), Food-101 (Bossard et al. (2014)) (third row), and CIFAR-100 (Krizhevsky et al. (2009)) (last row) datasets.

initial prediction and the image description generated by the LLM (Gemini Team Google (2023)) – we utilize a sliding kernel with a stride of one word. We mask out words that match the kernel and quantify their importance in our final prediction. As shown in Figure 5, the three inputs collectively contribute to predicting the final class label. In some cases, one or two inputs exhibit a higher level of influence than the others, as demonstrated in the first, second, and third examples.

It is worth mentioning that, while Gemini's initial predictions do not match the ground-truth class in the first and second examples, the predictions are contextually sensible. In the first example, Gemini's prediction was 'tunnel' which, while not directly matching any class label in the Places dataset (Zhou et al. (2017)), conceptually aligns with the displayed 'catacomb' image as an underground passage. Similarly, in the second example, Gemini's initial prediction was 'Snoopy', which corresponds to the character drawn on the pen shown in the input image. However, 'Snoopy' is not one of the ImageNet (Deng et al. (2009)) class labels and the correct class of the shown image in second row of Figure 5 is 'ballpoint pen'. This behavior of LLMs is the reason we cannot use them directly as image classifiers, because they sometimes do not restrict the output class to the provided list of target classes. However, such behavior might be beneficial to other classification tasks that are not restricted to a specific set of classes. Additional examples are provided in the supplemental materials (Appendix B).

## 3.2 ABLATION STUDIES

We conducted a series of ablation studies to explore different versions of our method and investigate the impact of each feature, different fusion approaches, and different cross-modal embedding mod-

Table 2: Ablation study on the impact of features used by our method on the classification accuracy. DF refers to the description feature, PF refers to the prediction feature, and IF refers to the image feature. In all datasets, we employed the best class feature as indicated in Table 1. Specifically, we utilized the combined class feature for all datasets except for Caltech-101 (Fei-Fei et al. (2004)), where we opted for the class description feature. The best results are highlighted in **bold**.

| | Dataset | | | | | | | | | |
|---|---|---|---|---|---|---|---|---|---|---|
| **Method** | ImageNet | C-10 | C-100 | Food | SUN | Cars | DTD | Caltech | Pets | Places |
| Ours (DF) | 58.6 | 90.1 | 64.5 | 82.7 | 49.0 | 65.4 | 49.8 | 83.7 | 48.3 | 30.1 |
| Ours (PF) | 55.7 | **94.6** | 73.2 | 89.6 | 60.9 | 70.8 | 57.7 | 89.0 | 87.1 | 36.1 |
| Ours (DF and PF) | 64.5 | 94.4 | **74.0** | 89.8 | 61.6 | 71.6 | 57.7 | 89.3 | 87.5 | 37.0 |
| Ours (DF and IF) | 70.7 | 90.4 | 64.0 | 90.8 | 67.9 | 71.4 | 52.6 | 90.9 | 85.5 | 42.1 |
| Ours (PF and IF) | 71.6 | 92.0 | 67.2 | 92.2 | 69.7 | 74.1 | 56.4 | 91.1 | 90.6 | 42.7 |
| Ours (DF, PF, and IF) | **73.4** | 93.4 | 70.2 | **93.0** | **70.6** | **76.6** | **58.0** | **91.3** | **90.9** | **43.4** |

Table 3: Ablation study on various fusion approaches using 5,000 images randomly selected from the ImageNet dataset (Deng et al. (2009)). The best results are highlighted in **bold**.

| | CLIP model (Radford et al. (2021)) | | |
|---|---|---|---|
| **Fusion Approach** | ViT-L/14 | ViT-B/32 | ViT-B/16 |
| Max similarity | 57.6 | 57.8 | 57.9 |
| Avg similarity | 66.1 | 65.5 | 65.8 |
| Avg feature | **72.5** | 65.7 | 68.5 |

els. Table 2 presents the results of our method using solely the encoded image description, referred to as the description feature (DF), as our input. We also report the results obtained by using encoded initial predictions, termed as the prediction feature (PF), as our input, as well as using both DF and PF concurrently as inputs. Additionally, Table 2 shows the results of employing image feature (IF) alongside DF or PF as inputs, and finally, we present the results when leveraging all available inputs – specifically, DF, PF, and IF.

From the results in Table 2, it is clear that incorporating all three features (DF, PF, and IF) yields the best performance across most datasets, except for the CIFAR datasets (Krizhevsky et al. (2009)). This discrepancy may arise from the low resolution of CIFAR images (originally 32×32), where utilizing the IF may degrade accuracy compared to using only DF and PF.

Table 3 shows the results of our second set of ablation studies, where we report the results for 5,000 randomly selected images from the ImageNet dataset (Deng et al. (2009)). We explore the use of different cross-modal embedding models (CLIP [ViT-L/14], CLIP [ViT-B/32], and CLIP [ViT-B/16]) (Radford et al. (2021)), and additionally investigate different fusion approaches. Rather than using the mean feature vector of our input features (DF, PF, and IF), we calculated the similarity between each input feature separately and the dataset class label features. Subsequently, we fused the similarity scores to generate a single similarity score for each class label in each dataset. We explored two fusion methods: averaging and taking the maximum for each class label. As shown, averaging our three features (IF, DF, PF) yields the best results.

## 4 COMPUTATION RESOURCES

Our method relies on a multimodal LLM and cross-modal encoders. The cross-modal encoding takes around 15 ms to encode an image or text on an NVIDIA V100 GPU, while the LLM can be accessed through: 1) Cloud API calls, which do not require local resources to load the model, or 2) loading the model locally for processing, which requires an estimated 16 GPUs/TPUs with approximately 256 GB of memory. Each LLM query takes roughly 700 ms to process. The LLM model is the most intensive operation (as discussed in Section 5), but it can be accelerated using multi-threading.

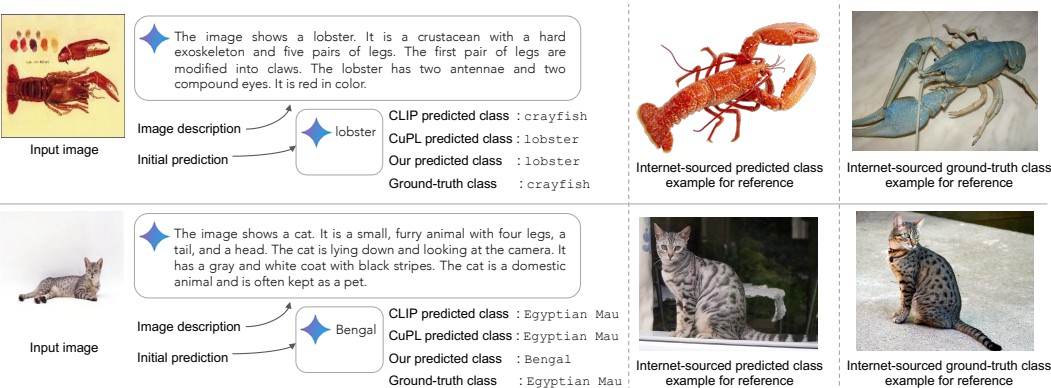

Figure 6: Failure examples of our method, where the initial prediction (and the image description in the first example) adversely influenced our final decision. Results are shown for the Caltech-101 dataset (Fei-Fei et al. (2004)) (first row) and the Pets dataset (Parkhi et al. (2012)) (second row).

## 5 LIMITATIONS

Our method introduces a new approach by leveraging multimodal LLMs to enhance the accuracy of zero-shot image classification. However, it is important to acknowledge that there are still some limitations inherent in our proposed method. Since our method relies on multiple queries to a multimodal LLM to generate the required features (i.e., DF and PF), there may be potential constraints when running on devices with limited computational power, and it may consume more time compared to other methods. Nevertheless, we believe that advancements in LLMs will lead to models that can run efficiently on lower computational power. This would enable broader accessibility and applicability of such models, such as Gemini (Gemini Team Google (2023)), GPT (Brown et al. (2020)), and LLaMA (Touvron et al. (2023)).

Our method fails in some cases. Figure 6 shows examples of failure cases, where our method misclassify the input image. While the initial prediction and image description features generally enhance classification accuracy, as demonstrated in Table 2, they can sometimes lead to misclassifications. In the first example in Figure 6, both the image description and initial prediction suggest that the image show a 'lobster', whereas it actually shows a 'crayfish'. Similarly, in the second example, the image description lacks specific features of the cat, while the initial prediction suggests the 'Bengal' class label, whereas the actual class label is 'Egyptian Mau'.

## 6 CONCLUSION

In this work, we introduced a zero-shot image classification method that relies on multimodal large language models (LLMs). Our approach involves using a multimodal LLM to describe the input testing image and make an initial class prediction based on input testing image and the target class label names. Subsequently, we fuse the encoded features of the image description, initial LLM's prediction, and input testing image to retrieve similar encoded features to class labels from the target dataset. Our method is straightforward and easy to implement, resulting in significant improvements in zero-shot classification accuracy when compared with prior methods in this domain.

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

## A ADDITIONAL DETAILS

In the main paper, we presented our method for zero-shot image classification. The inference process of our method is concisely described in Algorithm 1. As part of our method, we employed a set of prompts. Table 4 shows the prompts used for each step discussed in the main paper that employs the LLM (Gemini Pro (Gemini Team Google (2023))). Specifically, we detail the prompts used to: 1) conduct zero-shot image classification with Gemini Pro (Gemini Team Google (2023)), 2) describe a given testing image, and 3) generate class labels descriptions.

The class descriptions were generated using five prompts, as shown in Table 4, with 10 responses generated for each prompt and class, resulting in 50 class descriptions per class label. To encourage

Table 4: Details of prompts utilized in our work. Each row represents one query task to the LLM. For instance, 'image classification' indicates the utilization of LLM to conduct initial zero-shot image classification, which serves as one of the features in our method. The {classes} variable refers to the class labels of the dataset. The {predicted_class} refers to Gemini Pro's output of the image classification prompt. The {class_label} variable denotes one of the class labels in the given dataset.

| Task | Prompt |
|---|---|
| **Image classification** | `You are given an image and a list of class labels. Classify the image given the class labels. Answer using a single word if possible. Here are the class labels: {classes}` |
| **Image description** | `What do you see? Describe any object precisely, including its type or class.` |
| **Class description** | `1. Describe what a {class_label} looks like in one or two sentences.`
`2. How can you identify a {class_label} in one or two sentences?`
`3. What does a {class_label} look like? Respond with one or two sentences.`
`4. Describe an image from the internet of a {class_label}. Respond with one or two sentences.`
`5. A short caption of an image of a {class_label}:` |

Table 5: Additional results on 5,000 images from the ImageNet dataset (Deng et al. (2009)). Best result is highlighted in yellow

|  | L/14 | B/16 | B/32 | DistilBERT | RoBERTa | ROUGE-N-F1 | ROUGE-F1 | Ours |
|---|---|---|---|---|---|---|---|---|
| Top-1 | 52.1 | 52.0 | 55.6 | 43.7 | 28.8 | 54.8 | 54.9 | **70.2** |

diversity in Gemini Pro's responses, we set the temperature parameter to a high value of 0.99, as done in Pratt et al. (2023). An example of implementing our method, including both classifier construction and the inference process, is shown in Code 1.

In Section 3.1, we visualize examples that highlight the important parts of the inputs contributing to the final predicted class label. In the main paper, we described the approach of sequentially masking out patches from the image and comparing the predicted class with the prediction obtained using the entire unmasked image. Similarly, we follow the same approach for text input by sliding a kernel, masking out words, and comparing the predicted class with our original prediction using inputs without any masking. We used a 2D kernel of size 50×50 pixels with a stride of 10 pixels. If there are no highlighted regions in the image due to the small size of the kernel, we enlarge it by 50 until we reach a kernel size of 200×200 pixels.

For the text kernel, we start with a kernel width of 3 words. If none of the words are highlighted, we reduce it by 1 until we use a 1-word kernel sliding over the text. Each prediction was made using the three inputs: the image, initial prediction, and image description, with one of them having masked out patches or words.

## B ADDITIONAL RESULTS

In this section, we provide supplementary results to those presented in the main paper. Figure 7 shows the confusion matrix for CLIP (ViT-L/14) (Radford et al. (2021)) and our method across two datasets (Caltech-101 (Fei-Fei et al. (2004)) and CIFAR-100 (Krizhevsky et al. (2009))). The shown

---

**Algorithm 1** Performs zero-shot image classification.

---

**Input:** Image $\mathbf{X}$, class labels $\{l_i\}_{i=1}^{m}$, class label feature matrix $\mathbf{M}$ (Equation 1), multimodal LLM $g$, cross-modal encoders $f_i$ & $f_t$, initial class prediction prompt $p_{\mathtt{c}}$, image description prompt $p_{\mathtt{d}}$

$\quad\quad \widetilde{\mathbf{X}}_{\mathtt{if}} = f_i\left(\mathbf{X}\right)$ $\hfill \triangleright$ Image feature

$\quad\quad \mathbf{X}_{\mathtt{if}} = \widetilde{\mathbf{X}}_{\mathtt{if}}/\|\widetilde{\mathbf{X}}_{\mathtt{if}}\|$ $\hfill \triangleright$ Vector normalization

$\quad\quad \widetilde{\mathbf{X}}_{\mathtt{df}} = \left(f_t \circ g\right)\left(\mathbf{X}, p_{\mathtt{d}}\right)$ $\hfill \triangleright$ Image description feature

$\quad\quad \mathbf{X}_{\mathtt{df}} = \widetilde{\mathbf{X}}_{\mathtt{df}}/\|\widetilde{\mathbf{X}}_{\mathtt{df}}\|$ $\hfill \triangleright$ Vector normalization

$\quad\quad \widetilde{\mathbf{X}}_{\mathtt{pf}} = \left(f_t \circ g\right)\left(\mathbf{X}, p_{\mathtt{c}}\right)$ $\hfill \triangleright$ Initial class prediction feature

$\quad\quad \mathbf{X}_{\mathtt{pf}} = \widetilde{\mathbf{X}}_{\mathtt{pf}}/\|\widetilde{\mathbf{X}}_{\mathtt{pf}}\|$ $\hfill \triangleright$ Vector normalization

$\quad\quad \widetilde{\mathbf{X}}_{\mathtt{q}} = \mathbf{X}_{\mathtt{if}} + \mathbf{X}_{\mathtt{df}} + \mathbf{X}_{\mathtt{pf}}$ $\hfill \triangleright$ Fused feature

$\quad\quad \mathbf{X}_{\mathtt{q}} = \widetilde{\mathbf{X}}_{\mathtt{q}}/\|\widetilde{\mathbf{X}}_{\mathtt{q}}\|$ $\hfill \triangleright$ Vector normalization

$\quad\quad \mathbf{W} = \mathbf{X}_{\mathtt{q}}^{T}\mathbf{M}$ $\hfill \triangleright$ Similarity scores

$\quad\quad x \leftarrow \mathrm{argmax}\left(\mathbf{W}\right)$ $\hfill \triangleright$ Predicted class index

**Output:** Predicted class label $l_x$ of input image

---

Table 6: Top-5 classification accuracy of CLIP (Radford et al. (2021)), CuPL (Pratt et al. (2023)), and our method on the following datasets: ImageNet (Deng et al. (2009)), CIFAR-10 (C-10) (Krizhevsky et al. (2009)), CIFAR-100 (C-100) (Krizhevsky et al. (2009)), Food-101 (Bossard et al. (2014)), SUN397 (Xiao et al. (2010; 2016)), Cars (Krause et al. (2013)), DTD (Cimpoi et al. (2014)), Caltech-101 (Fei-Fei et al. (2004)), Pets (Parkhi et al. (2012)), and Places (Zhou et al. (2017)). We report our results with the following class label features: 1) class descriptions, 2) class labels, 3) the template "A photo of {class}", and 4) combined features of (1-3). The best results are highlighted in `yellow`.

| Method | ImageNet | C-10 | C-100 | Food | SUN | Cars | DTD | Caltech | Pets | Places |
|---|---|---|---|---|---|---|---|---|---|---|
| | | | | | | | | | | Dataset |
| CLIP (ViT-L/14) (Radford et al. (2021)) | 88.4 | 98.5 | 77.0 | 97.8 | 89.1 | 93.7 | 72.8 | 95.2 | 96.6 | 64.5 |
| CuPL (Pratt et al. (2023)) | 91.0 | 98.1 | 79.3 | 98.3 | 92.1 | 94.2 | 77.7 | 99.8 | 96.2 | 68.8 |
| Ours (class descriptions) | 92.7 | 99.3 | 85.4 | 98.9 | 93.8 | 97.6 | 81.0 | **99.9** | 96.9 | 70.7 |
| Ours (class labels) | 89.9 | 99.2 | 84.3 | 98.8 | 91.1 | 97.5 | 77.6 | 98.9 | 98.6 | 67.5 |
| Ours (template) | 89.8 | **99.6** | 84.8 | 97.9 | 90.1 | 97.5 | 77.2 | 96.6 | 97.3 | 65.3 |
| Ours (combined) | **93.0** | **99.6** | **88.5** | **99.0** | **94.4** | **97.9** | **83.8** | **99.9** | **99.5** | **70.9** |

results demonstrate that our method enhances classification accuracy and reduces misclassification rates.

In Table 2, encouraging results were demonstrated by utilizing the feature of initial prediction produced by the LLM (i.e., Gemini Pro (Gemini Team Google (2023))) for zero-shot image classification. Based on these results, one might argue for the direct utilization of Gemini's class prediction, aiming to match a specific class label from the dataset. However, in several cases, Gemini's response does not precisely match one of the class labels (as shown in Figure 5). For example, if a ground-truth class label is 'cat', Gemini's response might be 'The image class is cat'. This discrepancy motivated us to report results of using only Gemini prediction.

In this section, we present additional results from early experiments aimed at utilizing Gemini's predictions to precisely match one of the class labels in the given dataset. Specifically, we randomly selected 5,000 images from ImageNet (Deng et al. (2009)) for evaluation. While our method, as presented in the main paper, offers a practical way of utilizing Gemini's predictions, we also present the results of some alternative approaches aimed at precisely identifying one of the dataset class labels, rather than solely relying on the class prediction text generated by Gemini.

Table 5 show the results on the 5,000 images from ImageNet (Deng et al. (2009)) of our main method and alternatives that utilize Gemini's class prediction to conduct similarity matching with the target dataset class labels. Specifically, we report the results of encoding Gemini's class prediction using an open-vocabulary language model and measuring the similarity with the encoded class label features. Here, we show the results of using CLIP (ViT-L/14, ViT-B/32, ViT-B/16) (Radford et al. (2021)), DistilBERT (Sanh et al. (2019)), and RoBERTa (Liu et al. (2019)).

In addition, we explored classical text similarity metrics – namely, ROUGE-N-F1 and ROUGE-F1 (Lin (2004)) – rather than encoding both Gemini's prediction and class labels using an open-

```python
import tensorflow as tf
import cross_modal_encoder as encoder  # for example CLIP
import llm  # for example Gemini Pro
from fixed_prompts import classification_p, description_p, class_ps  # see Table 4.

def create_classifer(class_names, k=50):
  '''Constructs zero-shot image classifier.
  Args:
    class_names: A list of class names.
    k: Number of class descriptions to be generated by the LLM.
  Returns:
    A zero-shot image classification model.
  '''
  assert k >= len(class_ps)
  assert k % len(class_ps) == 0
  weights = []
  for class_name in class_names:
    class_name_feature = encoder.encode_text(class_name)
    template_feature = encoder.encode_text(f"A photo of {class_name}")
    llm_class_description = tf.zeros((1, encoder.output_feature_length))
    for _ in range(k // len(class_ps)):
      for class_p in class_ps:
        llm_class_feature = llm.process(class_p.format(class_name), temperature=0.99)
        llm_class_description += encoder.encode_text(llm_class_feature)
    llm_class_description /= k
    class_feature = class_name_feature + template_feature + llm_class_description
    normalized_class_feature = class_feature / tf.norm(class_feature)
    weights.append(tf.squeeze(normalized_class_feature))
  model = {"weights": tf.transpose(tf.convert_to_tensor(weights)),
           "class_names": class_names}
  return model

def classify(image, classifier):
  '''Performs zero-shot image classification.
  Args:
    image: Input testing image.
    classifier: A zero-shot classification model generated by create_classifier function.
  Returns:
    Predicted class name.
  '''
  image_feature = encoder.encode_image(image)
  image_feature /= tf.norm(image_feature)
  initial_prediction = llm.process([classification_p, image], temperature=0)
  prediction_feature = encoder.encode_text(initial_prediction)
  prediction_feature /= tf.norm(prediction_feature)
  image_description = llm.process([description_p, image], temperature=0)
  description_feature = encoder.encode_text(image_description)
  description_feature /= tf.norm(description_feature)
  query_feature = image_feature + prediction_feature + description_feature
  query_feature /= tf.norm(query_feature)
  index = tf.argmax(tf.linalg.matmul(query_feature, classifier["weights"]))
  return classifier["class_names"][index.numpy().squeeze()]
```

Code 1: Example Python implementation of our method. In this example, we utilize the combined class feature, as described in Section 3.

vocabulary encoding model. As shown in Table 5, our method, which utilizes Gemini's class prediction as one of the input features, achieves the best results when compared with the alternative approaches.

In the main paper, we reported the top-1 classification accuracy on several datasets (Deng et al. (2009); Fei-Fei et al. (2004); Bossard et al. (2014); Krause et al. (2013); Krizhevsky et al. (2009); Cimpoi et al. (2014); Zhou et al. (2017); Xiao et al. (2010; 2016); Parkhi et al. (2012)). Table 6 presents the top-5 classification accuracy of our method compared to prior work, while Table 7 shows the Cohen's kappa coefficient. As can be seen, our method achieves a notable improvement while remaining simple and easy to implement.

Lastly, Figure 8 shows additional visual examples, where we highlight the most significant contributors from input parts that influence the final predictions of our method.

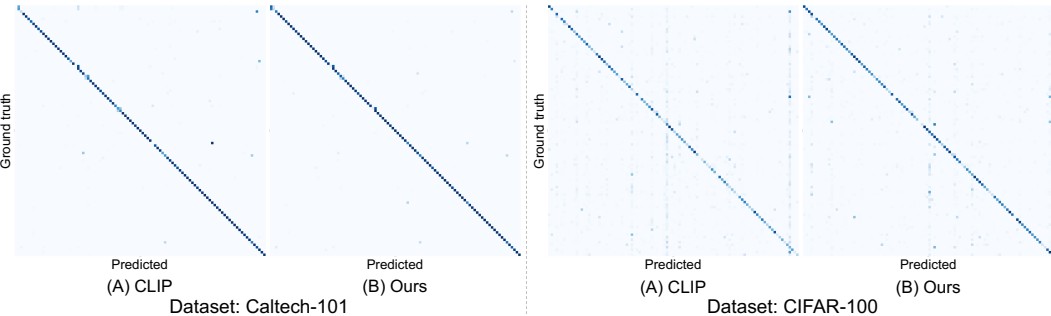

Figure 7: Confusion matrices for zero-shot image classification results of (A) CLIP (ViT-L/14) (Radford et al. (2021)) and (B) our method on the Caltech-101 (Fei-Fei et al. (2004)) and CIFAR-100 (Krizhevsky et al. (2009)) datasets.

Table 7: Cohen's Kappa score of CLIP (Radford et al. (2021)), CuPL (Pratt et al. (2023)), and our method on the following datasets: ImageNet (Deng et al. (2009)), CIFAR-10 (C-10) (Krizhevsky et al. (2009)), CIFAR-100 (C-100) (Krizhevsky et al. (2009)), Food-101 (Bossard et al. (2014)), SUN397 (Xiao et al. (2010; 2016)), Cars (Krause et al. (2013)), DTD (Cimpoi et al. (2014)), Caltech-101 (Fei-Fei et al. (2004)), Pets (Parkhi et al. (2012)), and Places (Zhou et al. (2017)). We report our results with the following class label features: 1) class descriptions, 2) class labels, 3) the template "A photo of {class}", and 4) combined features of (1-3). The best results are highlighted in **yellow** .

| | **Dataset** | | | | | | | | | |
|---|---|---|---|---|---|---|---|---|---|---|
| **Method** | ImageNet | C-10 | C-100 | Food | SUN | Cars | DTD | Caltech | Pets | Places |
| CLIP (ViT-L/14) (Radford et al. (2021)) | 0.651 | 0.862 | 0.539 | 0.867 | 0.611 | 0.656 | 0.452 | 0.830 | 0.835 | 0.372 |
| CuPL (Pratt et al. (2023)) | 0.665 | 0.851 | 0.573 | 0.889 | 0.652 | 0.637 | 0.480 | 0.902 | 0.795 | 0.395 |
| Ours (class descriptions) | 0.713 | 0.902 | 0.650 | 0.924 | 0.687 | 0.718 | 0.541 | **0.911** | 0.846 | 0.418 |
| Ours (class labels) | 0.699 | 0.909 | 0.650 | 0.920 | 0.667 | 0.742 | 0.522 | 0.882 | 0.897 | 0.411 |
| Ours (template) | 0.709 | 0.925 | 0.675 | 0.919 | 0.668 | 0.747 | 0.545 | 0.874 | 0.892 | 0.412 |
| Ours (combined) | **0.734** | **0.927** | **0.699** | **0.929** | **0.706** | **0.764** | **0.571** | 0.890 | **0.907** | **0.432** |

## C  BROADER IMPACT

Our work introduces a method for zero-shot image classification that leverages the power of multimodal large language models (LLMs) not only during the classifier model construction phase but also at inference time. We achieve this by generating comprehensive textual representations directly from input images. These representations are then combined with the input images for classification, resulting in a significant enhancement in accuracy.

Importantly, our approach eliminates the need for dataset-specific prompt engineering, as commonly required in prior approaches, thereby simplifying the implementation process and enhancing accessibility – effectively acting as a plug-and-play solution. By removing the requirement for dataset-specific customization, our method offers a straightforward and user-friendly approach to zero-shot image classification, making it more accessible to a broader range of users.

By demonstrating its effectiveness across diverse datasets,we illustrate the utility of our method for robust and generalizable real-world computer vision systems reliant on image classification, eliminating the need for dataset-specific training, tuning, or prompt engineering. This approach holds promise for simplifying the deployment of image classification systems and advancing the field of computer vision.

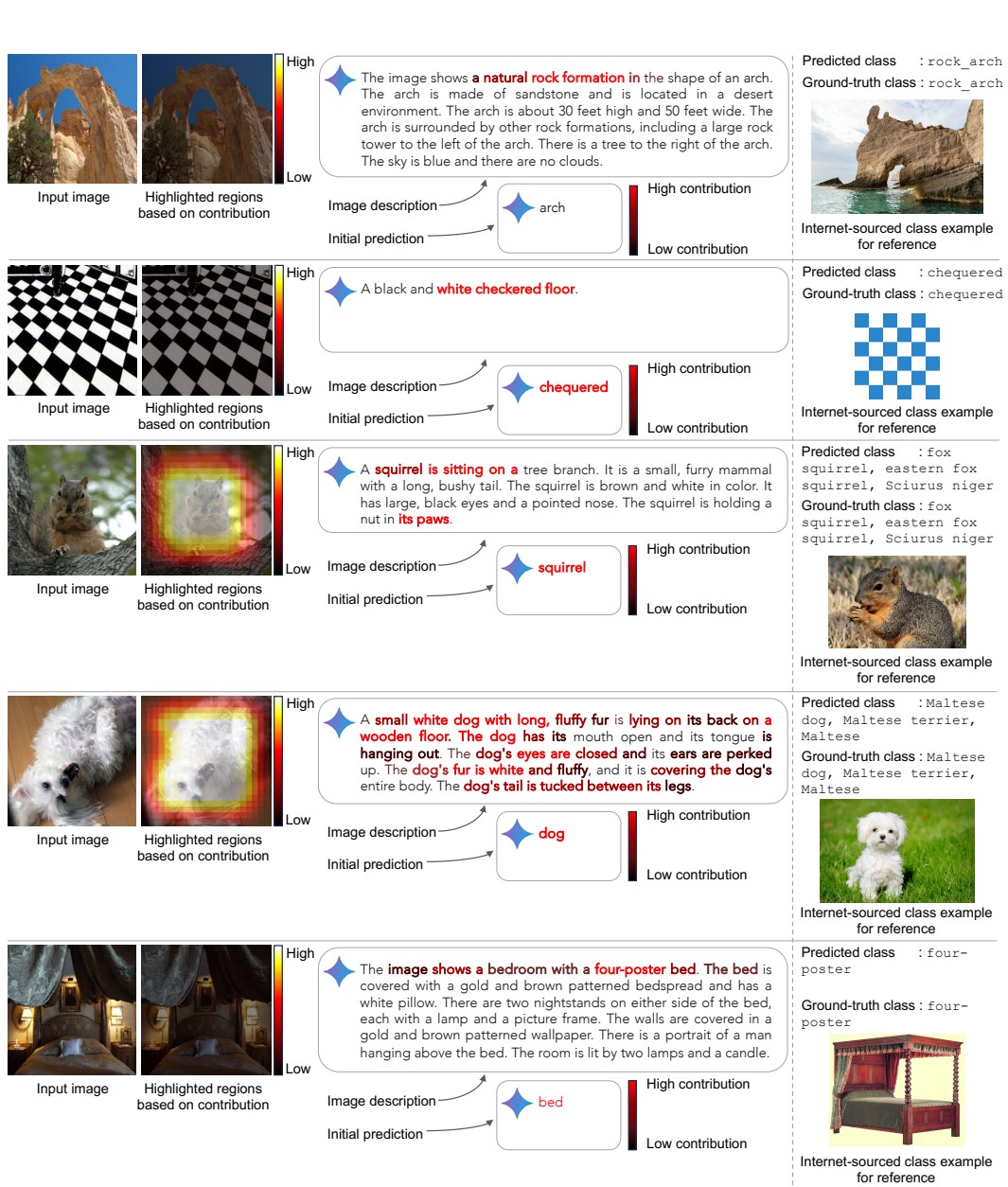

Figure 8: Additional examples demonstrating the influence of input data on final predictions. Examples are provided from the following datasets: SUN397 (Xiao et al. (2010; 2016)) (first row), DTD (Cimpoi et al. (2014)) (second row), and ImageNet (Deng et al. (2009)) (last three rows).

