# OpenReview forum: "What Do You See? Enhancing Zero-Shot Image Classification with Multimodal Large Language Models"
_ICLR.cc/2025/Conference — ICLR 2025 Conference Withdrawn Submission_

### Official Review · Reviewer_WY8P · 2024-10-20

**Soundness:** 2
**Presentation:** 2
**Contribution:** 2
**Rating:** 3
**Confidence:** 3

**Summary:**

The paper mainly focuses on performing zero-shot image classification tasks with multi-modal LLM. Specifically, in this paper, the authors propose a simple yet effective approach to first generate comprehensive textual representations from input images through multi-modal LLMs and then fuse the obtained features to perform zero-shot image classification with a linear classifier. The proposed method is free of prompt engineering. According to the reported results, the proposed method achieves good performance.

**Strengths:**

- The paper introduces MLLM to perform zero-shot image classification.
- The proposed method is prompt-free compared to previous works.
- The empirical results are good.

**Weaknesses:**

- The motivation of the paper is not clear. The concrete information is provided in Q1.
- The descriptions of the proposed method are confusing, see Section 2.
- According to the methodology part, the paper merely fuses the features of three sets of data. The novelty of the paper is extremely weak.

__Minor Concerns:__
- The legend in Fig. 2 is not well defined, which is a little bit confusing.
- The definition of $||\cdot||$ in Eq. (3) is incorrect.

__I have tried my best to read the paper and figure out the motivation and method proposed. However, I failed. The writing of the paper is bad and confusing. The proposed method seems naive. I don't think that this paper reach the bar of ICLR. Thus, I vote to reject.__

**Questions:**

1. The main motivation of the paper is that "relying solely on visual features of the input images during inference can limit accuracy". However, such a statement is not clear and convincing enough for me to understand the motivation of the paper. I am confused with such a motivation. Could you please make it clear?
2. As stated in the paper, the visual features of the input images are not sufficient enough to fully capture the nuances in the texture descriptions. To make the paper more solid, it would be better to provide some cases here and discuss which kind of features are competent features for the task.
3. It would be better to provide enough explanations for the notations in the proposed method part. For example,  what does $f_i$ and $f_t$ mean? Current statements are quite confusing.
4. Why is it necessary to generate additional textual-based inputs?
5. There are too many trivial equations in the paper, such as Eq. (2 - 9).

---

### Official Review · Reviewer_YH3x · 2024-10-29

**Soundness:** 2
**Presentation:** 2
**Contribution:** 2
**Rating:** 5
**Confidence:** 4

**Summary:**

The paper introduces a training-free method for improving zero-shot image classification of vision-language models (VLMs) using auxiliary multimodal large language models (MLLMs), i.e. Gemini Pro. With a hand-crafted, dataset-independent prompt, the authors use the MLLM to generate textual descriptions and initial predictions for the input image. The VLM's image and text encoders are then used to extract the input image and the generated text features. These features are fused by averaging. A linear classifier is built using the downstream class names. Classification logits are obtained by multiplying the fused features with the linear classifier. The method achieves an average accuracy gain of 4% across ten datasets.

**Strengths:**

- The paper is well-organized and clearly written, making the experimental setup easy to follow.
- A key strength of this method is that it eliminates the need for dataset-specific prompt engineering. This simplifies the pipeline and enhances its applicability across various datasets.
- The approach demonstrates significant improvements in zero-shot classification performance across ten benchmark datasets, particularly on ImageNet

**Weaknesses:**

- One major issue is the use of different backbone models compared to the baselines in the main comparisons (Table 1). The paper employs a CLIP ViT-L/14  for their method (line 243) but compares it to baselines using smaller models like CLIP ViT-B/32, ViT-B/16, and RN50 (and many more), which leads to an unfair advantage. Including smaller backbones for their method, would have enabled a more comprehensive evaluation.
- Additionally, the method’s reliance on Gemini Pro as a multimodal LLM raises concerns about accessibility and generalizability, since not all researchers have access to such powerful models.
- Furthermore, the paper lacks sufficient ablation studies to assess the contributions of different components, particularly the impact of the size of the VLM's backbone model and the role of the MLLM (i.e., the classification performance of Gemini Pro by itself).
- Finally, the greatest weakness is the additional computational cost of using this method in real-world scenarios compared to the baselines. The proposed dataset-independent prompt still requires knowing the training class names and remains image-dependent even at test time. It requires prompting the MLLM to obtain the zero-shot image classification and the description (two MLLM queries). Additionally, the reported results use 50 class descriptions for each label (line 593); these amount to 50,000 more queries (since ImageNet has 1,000 classes) to the MLLM for testing on ImageNet.

**Questions:**

- Q1. Have you considered reformulating your evaluation, running experiments with the same backbone model, and comparing the performance with and without your proposed method? Your method is a general plug-and-play approach that could enhance the classification performance of any VLM. I think that empirically showing a robust improvement on many backbones could strengthen your hypothesis.
- Q1bis. It is not clear in Table I which experiments have been recalculated and which backbones have been used for the compared baselines. For example, is CuPL using the same backbone?
- Q2. As a potential future direction, it might be interesting to explore more accessible alternatives to Gemini Pro, such as widely available MLLMs or to compare with other public VLMs. It could even be interesting to see if a simple and more efficient captioner of the input image could improve the performance.
- Q3. My main concern relies on the benefits and improvements arising solely from using Gemini Pro. It would be really insightful to see how Gemini Pro performs on its own for classification. Have you thought about sharing its standalone performance? It might give a clearer picture of its role in the overall method's success.

I would be happy to reconsider my opinion if the authors address my concerns, in particular regarding the unfair comparison using different backbones and the additional computational cost.

---

### Official Review · Reviewer_D7yG · 2024-11-02

**Soundness:** 2
**Presentation:** 2
**Contribution:** 2
**Rating:** 3
**Confidence:** 5

**Summary:**

This paper introduces a method for zero-shot image classification using multimodel LLM. The proposed method first generates comprehensive textual representations from input image, and then utilizes these method textural representations to generate fixed-dimensional features in a cross-modal embedding space, finally these features are used to zero-shot classification.

**Strengths:**

This is a complete work and proposes to enhance the classification accuracy via introducing richer textual information generated via LLMs.
The proposed method employs a simple and universal set of prompts and free from the task of dataset-specific prompt engineering.
The proposed method is experimentally validated.

**Weaknesses:**

The method introduced in this paper is quite simple. It simple combine the textual and visual inputs together to prompt the accuracy.
This work is lack of theoretical analysis. Why and how much the proposed method can enhance the accuracy?
The experimental section is lack of implementation details.

**Questions:**

The method introduced in this paper is quite simple. It simple combine the textual and visual inputs together to prompt the accuracy.
This work is lack of theoretical analysis. Why and how much the proposed method can enhance the accuracy?
The experimental section is lack of implementation details.

---

### Official Review · Reviewer_r9KY · 2024-11-02

**Soundness:** 3
**Presentation:** 3
**Contribution:** 2
**Rating:** 5
**Confidence:** 5

**Summary:**

This paper proposes to improve zero-shot classification by using multimodal LLMs. The proposed method computes additional textual embeddings for each image in the test set by prompting a multimodal LLM and converting the prompt to an embedding using CLIP's text encoder. The paper utilizes two types of prompts for the multimodal LLM -- one which involves generating a description of the image, and the other which involves classifying the image. Results are shown on various datasets where the method improves over the baselines considered.

**Strengths:**

The method proposed is simple but effective. The paper is well presented where the method is presented with clarity. Ablations of the method has also been conducted for various contributing factors. Combination of various prompting strategies to the CLIP model for classification and their effects is also a useful contribution.

**Weaknesses:**

Missing datasets : Some datasets which are utilized by previous methods such as CUB, FGVC Aircraft, EuroSAT have been overlooked. Is it the case that the multimodal LLM does not possess useful knowledge about these fine-grained datasets and thus cannot improve performance?

Missing baseline : A useful baseline in Table 1 would be the accuracy of the Multimodal LLM in classifying the images. This result will give an insight into the improvements that combining the multimodal LLM and CLIP is bringing.

Comparison to few shot techniques : The authors compare to some few-shot techniques such as CLIP-Adapter, but do not list the accuracies for all datasets. Also it is not clear that given they choose to compare to few-shot techniques, why they choose to not compare with CoOp[1], CoCoOP[2], VDT-Adapter[3], AdaptCLIPZS[4] which are zero-shot techniques trained on "base" classes of a dataset. The last two mentioned works also use LLM generated attributes similar to this method, but have been overlooked.

Computational Overload : Prompting a multimodal LLM twice for each image is highly resource intensive, which brings into question the motivation of using this method in a real-world scenario. Methods such as transduction[5,6], few-shot training[1,2], training on different classes of the same domain[1,2,3,4] seem to be a better choice, therefore comparison with such methods should be conducted.

[1] Zhou, K., Yang, J., Loy, C.C. and Liu, Z., 2022. Learning to prompt for vision-language models. International Journal of Computer Vision, 130(9), pp.2337-2348.

[2] Zhou, K., Yang, J., Loy, C.C. and Liu, Z., 2022. Conditional prompt learning for vision-language models. In Proceedings of the IEEE/CVF conference on computer vision and pattern recognition (pp. 16816-16825).

[3] Maniparambil, M., Vorster, C., Molloy, D., Murphy, N., McGuinness, K. and O'Connor, N.E., 2023. Enhancing clip with gpt-4: Harnessing visual descriptions as prompts. In Proceedings of the IEEE/CVF International Conference on Computer Vision (pp. 262-271).

[4] Saha, O., Van Horn, G. and Maji, S., 2024. Improved Zero-Shot Classification by Adapting VLMs with Text Descriptions. In Proceedings of the IEEE/CVF Conference on Computer Vision and Pattern Recognition (pp. 17542-17552).

[5] Martin, S., Huang, Y., Shakeri, F., Pesquet, J.C. and Ben Ayed, I., 2024. Transductive Zero-Shot and Few-Shot CLIP. In Proceedings of the IEEE/CVF Conference on Computer Vision and Pattern Recognition (pp. 28816-28826).

[6] Zanella, M., Gérin, B. and Ayed, I.B., 2024. Boosting Vision-Language Models with Transduction. arXiv preprint arXiv:2406.01837.

**Questions:**

I would appreciate clarifications on the choices described in the weaknesses above.

---

### Official Review · Reviewer_AkZm · 2024-11-02

**Soundness:** 3
**Presentation:** 3
**Contribution:** 3
**Rating:** 6
**Confidence:** 5

**Summary:**

This paper introduces a novel approach for using the multimodal capabilities of modern LLMs for enhancing zero-shot classification. The proposed method is straightforward in its formulation: instead of only relying on text prompts generated by an LLM for a given class label, use three different multimodal features (image, image description, and an initial prediction) to form a multimodal query. The combined features are then compared against the list of class label features to find the label with maximum similarity, like the CLIP model. Despite its simple formulation, the method shows significant improvement in zero-shot classification across multiple standard datasets for this task.

**Strengths:**

- The paper is written very well, effectively communicating the method description and experiments in a concise and easy to understand way.
- The proposed approach shows significant improvement on almost every dataset traditionally used for benchmarking zero-shot image classification.
- It appears to be the first work using multimodal capabilities of MLLMs for this task, which is a natural next step from text-based features used in previous LLM methods for this task.

**Weaknesses:**

- The discussion of the results could be improved. For example, in lines 413-420, the results of the ablation study in Table 3 are described. But there is no discussion to explain the results. Particularly, line 420 says averaging features leads to best performance, but it would be better if the authors also provide a likely reason for why other fusion methods do not work.
- In Table 2 caption, the authors state they used the combined class features for all dataset except Caltech where class description features were used. Again, no reason is provided for this decision.
- Table 2 could also benefit from an additional row showing only image feature (IF) input for a more complete picture.
- Line 74-75 implies that feature fusion is illustrated in Figure 1, but the corresponding figure does not show this. The wording of this paragraph should be corrected.
- Lines 32-33 state that recent approaches have utilized LLMs to generate prompts for target classes, but this sentence does not cite these recent approaches. Some examples could be I2MVFormer by Naeem et al, or FocusCLIP by Khan et al. The authors should either use these or any other references which are suitable here.
- I am not completely convinced by the reason provided in Lines 411-412 for why the CIFAR dataset show best performance without the image features.

**Questions:**

- What could be the reason for CIFAR-10 working best with predicted class features (when those predicted classes are not even guaranteed to be correct), but not working as well with either class description features or image features?
- How often does the failure case in Lines 462-468 observed? Can the authors provide any quantitative measurement for this?

---

### Official Review · Reviewer_ZH8V · 2024-11-03

**Soundness:** 2
**Presentation:** 2
**Contribution:** 1
**Rating:** 3
**Confidence:** 4

**Summary:**

The paper uses pretrained multimodal large language models (LLMs) in zero-shot recognition. It proposes to generate textual represenatation from input images by asking multimodal LLMs such as Gemini Pro. It combines the generated textual representations with features computed by CLIP towards the final zero-shot recognition results. Experiments are done on some standard benchmarks. The paper is limited in novelty and misses citations of relevant works.

**Strengths:**

- It is reasonable to utilize pretrained foundation models to solve downstream tasks.
- Leveraging generated textual description in visual recognition is a research theme in recent years.

**Weaknesses:**

- As the paper concerns about "enhancing zero-shot image classification with multimodal large language models" (as suggested in the title), it fails to compare other zero-shot image classification methods which especially leverage pretrained foundation models and their pretraining data, such as [R1-R3].

- The setting is a more serious issue. While datasets used in experiments are standard ones for zero-shot recognition in the literature, images in these datasets are collected on the web, from where one collected data to train the foundation models. Therefore, using output by a (commercial) foundation model such as Gemini Pro is worrisome as it might have information leakage problems. Therefore, a more rigorous setting is to use more highly-specialized tasks which can challenge foundation models.

- While experiments show that the proposed methods outperform the compared ones, it is natural to believe that the benefit is from the use of large pretrained foundation models, especially the commercial Gemini Pro by Google. It makes the paper lack of technical novelties and not ready to publish.

Citations to compare
- [R1] Learning customized visual models with retrieval-augmented knowledge. CVPR 2023
- [R2] Neural priming for sample efficient adaptation. NeurIPS 2023
- [R3] The Neglected Tails of Vision-Language Models. CVPR 2024

**Questions:**

Authors are encouraged to address the weaknesses in rebuttal/responses. Please refer to the weaknesses for details.

---

### Official Review · Reviewer_jUUr · 2024-11-03

**Soundness:** 3
**Presentation:** 4
**Contribution:** 3
**Rating:** 5
**Confidence:** 2

**Summary:**

This paper presents a novel method that leverages multimodal large language models (LLMs) to enhance zero-shot image classification performance. By generating rich textual descriptions for input images and fusing these representations with visual features in a cross-modal embedding space, the approach outperforms several benchmark methods in terms of accuracy. The method uses a consistent set of prompts across datasets, avoiding dataset-specific prompt engineering. The experiments indicate significant performance gains over prior zero-shot approaches, especially in datasets like ImageNet.

**Strengths:**

The proposed method introduces a multimodal LLM-based framework to improve zero-shot image classification by integrating image descriptions and initial class predictions, which is relatively novel and improves on the limitations of using visual features alone.

**Weaknesses:**

The method’s reliance on a multimodal LLM (e.g., Gemini) may restrict its flexibility, as it depends on the capabilities and availability of specific LLMs, which could become a bottleneck.

**Questions:**

1.While the paper provides some qualitative examples of misclassifications, a more comprehensive error analysis could offer insights into potential areas for further improvement. For instance, understanding when and why the textual features lead to incorrect classifications would be valuable.
2.Could other LLM models be used for comparison in addition to Gemini?

---

### Note · Authors · 2024-11-15

I have read and agree with the venue's withdrawal policy on behalf of myself and my co-authors.